# DEGauss: Defending Against Malicious 3D Editing for Gaussian Splatting

**Lingzhuang Meng[1], Mingwen Shao[2],[*] Yuanjian Qiao[3], Xiang Lv[1]**

[1] Shandong Key Laboratory of Intelligent Oil & Gas Industrial Software,
Qingdao Institute of Software, College of Computer Science and Technology,
China University of Petroleum (East China), China
[2] Artificial Intelligence Research Institute, Shenzhen University of Advanced Technology, China
[3] College of Computer Science (College of Software), Inner Mongolia University, China
lzhmeng1688@163.com, smw278@126.com, yjqiao58@163.com, lvxiang1997@126.com

## Abstract

3D editing with Gaussian splatting is exciting in creating realistic content, but it also poses abuse risks for generating malicious 3D content. Existing 2D defense approaches mainly focus on adding perturbations to single image to resist malicious image editing. However, there remain two limitations when applied directly to 3D scenes: (1) These methods fail to reflect 3D spatial correlations, thus protecting ineffectively under multiple viewpoints. (2) Such pixel-level perturbation is easily eliminated during the iterations of 3D editing, leading to failure of protection. To address the above issues, we propose a novel **D**efense framework against malicious 3D **E**diting for **Gauss**ian splatting (**DEGauss**) for robustly disrupting the trajectory of 3D editing in multi-views. Specifically, to enable the effectiveness of perturbation across various views, we devise a view-focal gradient fusion mechanism that dynamically emphasizes the contributions of the most challenging views to adaptively optimize 3D perturbations. Furthermore, we design a dual discrepancy optimization strategy that both maximize the semantic deviation and the edit direction deviation of the guidance conditions to stably disrupt the editing trajectory. Benefiting from the collaborative designs, our method achieves effective resistance to 3D editing from various views while preserving photorealistic rendering quality. Extensive experiments demonstrate that our DEGauss not only performs excellent defense in different scenes, but also exhibits strong generalization across various state-of-the-art 3D editing pipelines.

## 1 Introduction

Recent advances in scene editing with 3D Gaussian Splatting (3DGS) [17] have substantially enhanced the controllability and expressiveness scene manipulation driven by natural language prompts, largely powered by the integration of diffusion models with neural 3D representations [12, 39]. However, such convenience also brings significant security risks: anyone with access to a rendered 3D model can change identity, appearance, or contextual details without authorization [25, 22], which can lead to serious consequences such as identity deception, misinformation, or reputational damage [31, 40]. Therefore, it is imperative to develop effective protection method for 3D digital assets and personal portraits against unauthorized modifications, in order to prevent the spread of malicious content.

Unlike 2D editing that operates on single image [15, 38], 3D editing fundamentally involves manipulating spatially structured data [20, 35], which brings in unique properties such as multi-view

---

*Corresponding authors.

39th Conference on Neural Information Processing Systems (NeurIPS 2025).

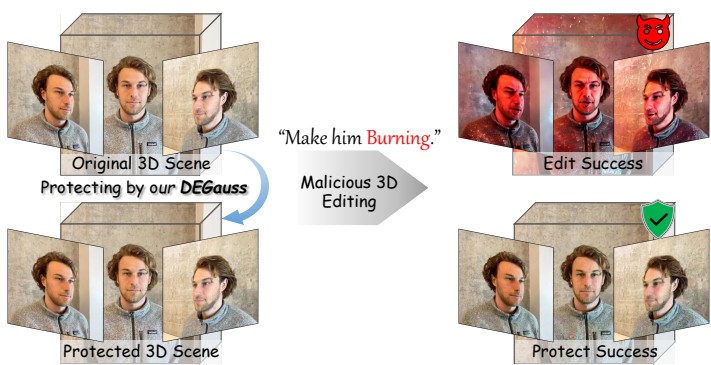

Figure 1: Malicious 3D editing and our defense. The 3D scenes protected by our DEGauss are able to disrupt the direction of malicious editing and preserve the original 3D appearance and structure.

consistency, structural integrity, and global iterative optimization. Existing studies [24, 3] on 3D editing based on Gaussian splatting and diffusion models have emphasized better semantic controllability, multi-view consistency, and editing flexibility. These methods aim to enable precise modifications of 3D content that align with user intent by local region mapping [7], while ensuring consistency across diverse viewpoints through the integration of semantic information from different viewpoints [4, 34]. Furthermore, recent works [18, 19] have increasingly focused on the editing efficiency and speed, enabling high-quality results with reduced computational cost and minimal guidance. Despite these significant advances, existing researches neglect the security risks posed by malicious use of 3D editing, which raises concerns about unauthorized modifications.

To defend against malicious editing on 2D images, various adversarial-based defense methods [22, 14, 26] have been proposed to disrupt the editing capabilities of diffusion models through embedded imperceptible perturbations. For example, AdvDM [22], Mist [21] and PhotoGuard [30] optimize the perturbations by maximizing the noise prediction loss of the diffusion model, thereby inducing significant degradation in downstream editing results. Posterior Collapse Attack [10] achieves attacks in latent space by inducing a posterior collapse in the VAE encoder, thus disrupting a variety of editing methods based on latent diffusion models. In addition, Semantic Attack [23] and AdvPaint [16] degrade editing quality by manipulating the self-attention and cross-attention mechanisms within the diffusion process, leading to misaligned attention maps and impaired semantic alignment. However, the aforementioned 2D defense methods are inherently unsuitable for 3D scenarios due to two main limitations: (1) Their 2D perturbations failed to reflect 3D spatial correlations, leading to poor protection across varying viewpoints. (2) Such perturbations are also easily erased during iterative optimization of 3D editing, resulting in defense failure.

To tackle the above challenges, we propose a tailored **D**efense framework against malicious 3D **E**diting for **Gauss**ian splatting (**DEGauss**), which stably disrupts the multi-view editing trajectory to suppress unauthorized 3D modifications, as shown in Fig. 1. Specifically, to ensure the effectiveness of perturbations across different viewpoints, we design a view-focal gradient fusion mechanism that dynamically emphasizes the gradient of the most challenging viewpoints in broadly sampled cameras, thereby adaptively optimizing the 3D perturbations under varying viewpoints. Furthermore, we devise a dual discrepancy optimization strategy that simultaneously maximizes the semantic deviation and the directional bias of the guidance condition, therefore enabling the 3D perturbation to robustly disrupt editing trajectories throughout the iterative optimization. With the support of these two dedicated designs, our method achieves strong resistance to multi-view editing, while preserving high-fidelity rendering with photorealistic visual details. Extensive experiments illustrate that our DEGauss effectively prevents malicious 3D editing across diverse scenes and exhibits promising generalization across a variety of 3D editing frameworks.

The main contributions of this paper are as follows:

- We propose DEGauss, a noval defense framework against malicious 3D editing for Gaussian splatting that actively disturbs the editing trajectory from multiple viewpoints. To the best of our knowledge, this is the first framework for defense against 3D malicious editing.

- A view-focus gradient fusion mechanism is designed to adaptively update 3D perturbations by emphasizing challenging views, thereby enhancing defense effectiveness in multi-views.

- We devise a dual-discrepancy optimization strategy that amplifies semantic divergence and editing directions errors in iterative optimization, stably disrupting the editing trajectory.

- Experiments on several datasets and state-of-the-art 3DGS editing schemes demonstrate the effectiveness and generalization of our DEGauss.

## 2 Related Works

**3D Editing with Gaussian Splatting.** Recent researches on 3D editing with 3DGS have focused almost on improving the fidelity [4, 18, 33], controllability [39, 7, 12, 28], and efficiency [18, 19]. For instance, to achieve controllable editing, GaussianEditor [7] introduces a dynamic semantic tracking strategy for precise localization by back-projecting 2D segmentation masks into 3D Gaussians. To ensure multi-view consistency, DGE [4] integrates spatio-temporal self-attention and epipolar constraints to achieve effective fusion of edits across views, while GaussCtrl [34] leverages geometric constraints on depth maps and cross-view attention mechanism to align edits across different views. To improve editing efficiency, ProEdit [3] adopts subtask-based optimization to gradually realize complex edits with better stability. On the contrary, DreamCatalyst [18] accelerates convergence via improved inversion and loss design, while EditSplat [19] further enhances speed through attention-weighted pruning and hierarchical densification. Despite the impressive progression of 3D editing, there are also raised concerns about its potential misuse to maliciously modify private 3D assets. Aiming at this problem, we propose DEGauss, the first defense framework against malicious 3D editing for protecting digital assets from arbitrary manipulation by unauthorized editor.

**Defenses against Malicious Image Editing.** To defend against malicious images editing, a series of adversarial-based defense methods [22, 6, 5, 14, 26] have been proposed to disrupt the editing process by exploiting imperceptible pixel-space perturbations. Among them, AdvDM [22] introduces a systematic theoretical framework that maximizes denoising loss of the diffusion models [29, 13], thereby generating perturbations to hinder feature extraction. On this basis, Mist [21] further combines semantic and texture losses to improve transferability of the perturbations, while DiffusionGuard [8] prevents the synthesis in sensitive regions by interfering with the early stages of the denoising process. In addition, SDS [36] accelerates optimization and reduces consumption through score distillation sampling, while Posterior Collapse Attack [10] induces potential spatial collapse by perturbing the VAE encoder in the latent diffusion model, thereby significantly disrupting the semantics of the encoding. Recently, Semantic Attack [23] and AdvPaint [16] perturb the text-image cross-attention or image self-attention in the diffusion model to distract the model attention to the specific region, and both of them showing better performance than conventional approaches. These defense schemes are investigated from the output space to the model structure to construct a more defense system against malicious editing of 2D images.

However, the above methods pose intrinsic challenges when applied directly to 3D scene protection, as they are specifically designed for single-view and single-pass 2D editing. On the one hand, the perturbations generated by 2D defense are unable to reflect 3D spatial relationships, leading to insufficient effectiveness across multiple viewpoints. On the other hand, simple perturbations optimized for single-pass editing are typically eliminated during the iterative optimization in 3D editing, resulting in failure of protection. In contrast, we propose DEGauss, a specialized defense framework against malicious 3D editing, which combines view-focal gradient fusion with dual-discrepancy optimization to robustly disrupt multi-view editing results during iterations of 3D editing.

## 3 Methodology

### 3.1 Preliminary

**3D Editing with Gaussian Splatting.** Existing mainstream of 3D editing with 3DGS employ pre-trained 2D diffusion models (e.g., InstructPix2Pix [2], ControlNet [37]) as powerful generative priors to supervise the editing results. Within these framework, text prompts are used to guide diffusion-based image editing for multiple viewpoints. The resulting edited images serve as supervision to update the Gaussian parameters $\Theta = \{\mu, \Sigma, \alpha, c\}$, where $\mu$ denotes the 3D center position, $\Sigma$

represents the covariance matrix controlling shape, $\alpha$ is the opacity, and $c$ defines the RGB color. The updated 3D representation is iteratively rendered to generate a new image for further editing.

Formally, given a constructed scene $G$, the goal of the editor is to update the Gaussian parameters $\Theta$ so that its rendered images from all viewpoints match the edited images according to the text prompt $y_{\text{text}}$. This update is performed iteratively:

$$\Theta = \arg\min_{\Theta} \mathbb{E}_{k \in \mathcal{U}(0,K), v \in V} \mathcal{L}_{edit}(I_v^k, D_\phi(I_v^k, I_v, y_{\text{text}})), \tag{1}$$

where $I_v^k = R_v(G^k)$ denotes the $k$-th round of iterations and the $v$-th view of the rendered image, $I_v$ represents the initial rendered image being used as image guidance to preserve the original information. $D_\phi$ denotes the diffusion model, and $\mathcal{L}_{edit}$ is the loss of rendering, which is commonly used to align the appearance of rendering (e.g., $L_1$, LPIPS) or geometric depth.

**Defense against Malicious Image Editing.** To defend against malicious image editing, existing researches has proposed adversarial-based methods to disrupt the editing process of diffusion models. The core principle of these approaches is to introduce a small-magnitude perturbation $\delta$ to the input image $x$ to make the diffusion model deviate from the original denoising process, thus affecting the normal editing operation. It can be formalized as:

$$\delta := \arg\min_{\delta} p(x + \delta), \quad s.t. \|\delta\| < \xi, \tag{2}$$

where $p(x)$ denotes the probability distribution of the image $x$ generated by the diffusion model and $\xi$ denotes the bound limit of the perturbation. To solve Eq. (2), AdvDM [22] represents the solution of the defense perturbation $\delta$ uniformly according to Monte Carlo as follows:

$$\delta = \arg\max_{\|\delta\| < \xi} \mathbb{E}_{t \sim \mathcal{U}(1,T), \epsilon \sim \mathcal{N}(0,\mathbf{I})} \|\epsilon_\theta(x_t, x + \delta, y_{\text{text}}, t) - \epsilon\|_2^2, \tag{3}$$

where $t \sim \mathcal{U}(1, T)$ denotes the number of denoising steps in the diffusion model, $\epsilon \sim \mathcal{N}(0, \mathbf{I})$ denotes normally distributed noise, and $\epsilon_\theta$ is the denoising network. By encouraging the noise predictions to deviate from the correct values at all time steps $t$, the denoising results of the diffusion model can be actively perturbed and ultimately achieve a robust defense against malicious image editing.

### 3.2 Problem Definition

In this paper, we present a defense against malicious 3D editing, aiming to prevent unauthorized modifications to 3D scenes performed by diffusion model-based editors. For a scene $G$ constructed by 3DGS, let $p_\phi(G_{\text{tar}}|G, y_{\text{text}})$ is the conditional probability distribution of the target 3D scene $G_{\text{tar}}$ produced by the diffusion editor (parameterized by $\phi$) given a text prompt $T$. We define the defense against malicious 3D editing as a probabilistic optimization problem.

**Definition 3.1 (Defense against malicious 3D editing).** Given a 3D editor, there exists 3D perturbation $\Delta$ added to the original scene $G$ that minimize the conditional probability of generating a valid edit result at text prompt $y_{\text{text}}$. The perturbation $\Delta$ is given by the following equation:

$$\Delta := \arg\min_{\Delta} p_\phi(G_{\text{tar}}|G + \Delta, y_{\text{text}}), \quad s.t. \quad \|\Delta\| < \xi. \tag{4}$$

Our goal is not limited to dissimilarity to a specific target distribution $G_{\text{tar}}$, but to reduce the probability of similarity to any edit distribution. Thus, the above equation can be written as:

$$\Delta := \arg\max_{\|\Delta\| < \xi} \mathbb{E}_{G_{\text{tar}} \sim p_\phi(\cdot | G, y_{\text{text}})}[-\log p_\phi(G_{\text{tar}}|G + \Delta, y_{\text{text}})]. \tag{5}$$

Combined with the optimization objective for 3D editing given in Eq. (1), we can reinterpret the defense problem as a maximization of the editing loss, as follows:

$$\Delta := \arg\max_{\|\Delta\| < \xi} \mathbb{E}_{k,v} \mathcal{L}_{\text{edit}}(I_v^k, D_\phi(I_v^k, R_v(G + \Delta), y_{\text{text}})). \tag{6}$$

However, this rendering loss is essentially an iterative fitting process to the explicit model, aiming to align the rendered image with the target edited image produced by the diffusion model. It is not possible to optimize the perturbation by maximizing this render loss $\mathcal{L}_{\text{edit}}$. Therefore, we turn to replace the original editing loss with a differentiable diffusion loss, which drives the 3D edit to fit to

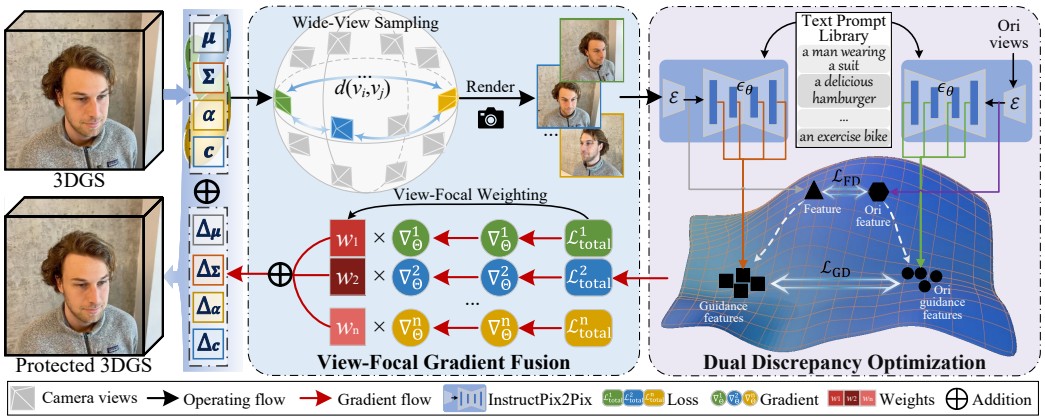

Figure 2: Overview of the proposed DEGauss. Firstly, a wide-viewpoint sampling is performed to obtain a rendered image of the 3D scene, and then feature discrepancy loss and guidance discrepancy loss are computed between the rendered and original view images. These losses from multiple viewpoints are subsequently fused using a view-focal weighting to guide perturbation optimization. The perturbation $\Delta = \{\Delta_\mu, \Delta_\epsilon, \Delta_\alpha, \Delta_c\}$ is initialized to zero and optimized in iterations.

the wrong edit direction by making the diffusion process in multiple views deviate from the original edit direction. This objective is formally defined as follows:

$$\min_{\|\Delta\|<\xi} \mathbb{E}_{v\in V} \mathcal{L}_{\text{diff}}(\mathcal{R}_v(G+\Delta), y_{\text{text}}) \tag{7}$$

$$= \max_{\|\Delta\|<\xi} \mathbb{E}_{k,v,t,\epsilon} \left\| \epsilon_\theta(I_v^k, \mathcal{R}_v(G+\Delta), y_{\text{text}}, t) - \epsilon \right\|_2^2. \tag{8}$$

Although Eq. (8) constructs theoretically reasonable attack targets based on the noise prediction of the diffusion model, it results in limited direct optimization due to the fact that the denoiser of the diffusion model is extremely robust to perturbations and the gradient is weak and unstable [36]. Therefore, we propose two loss terms, Feature Discrepancy (FD) and Guidance Discrepancy (GD), which directly measure the semantic differences and bias in the guidance direction in latent space. These contribute to more stable convergence and better gradient propagation.

$$\Delta := \arg\max_{\|\Delta\|<\xi} \mathbb{E}_{k,v,t,\epsilon} \underbrace{\left\| \varepsilon(\mathcal{R}_v(G+\Delta)) - \varepsilon(I_v) \right\|_2^2}_{\text{Feature Discrepancy}}$$
$$+ \underbrace{\left\| \epsilon_\theta(I_v^k, \mathcal{R}_v(G+\Delta), y_{\text{text}}, t) - \epsilon_\theta(I_v', I_v, y_{\text{text}}, t) \right\|_2^2}_{\text{Guidance Discrepancy}}, \tag{9}$$

where $\varepsilon$ represents the image encoder. The purpose of FD is to drive the protected view away from the original feature in latent space and increase the uncertainty of the editor. While the purpose of the GD is to alter the guidance direction of the diffusion model to deviate from the original trajectory, weakening its semantic understanding of $y_{\text{text}}$ and editing behavior.

Following the above analysis, we propose DEGauss, a dedicated framework that introduces targeted perturbations to disrupt editing across multiple views, as illustrated in Fig. 2. Our DEGauss consists of two key components: a view-focal gradient fusion mechanism and a dual-discrepancy optimization strategy. These components work collaboratively to optimize 3D spatial perturbations that reliably disrupt multi-view editing throughout the iterative process.

### 3.3 Dual Discrepancy Optimization

Based on Eq. (9), we devise a dual discrepancy optimization strategy that provides supervision from both the semantic feature and guidance direction in the latent space. Specifically, this module contains two dedicated loss functions: feature discrepancy loss and guidance discrepancy loss, which jointly guide the optimization of the perturbations, thus deviating editing results from target semantics.

**Feature Discrepancy Loss.** The feature discrepancy loss is used to measure the consistency between the original views and the perturbed 3D views in the latent space, and we utilize the encoder of pre-trained InstructPix2Pix [2] model to extract the feature representations of the rendered image and the original image, which can be represented as:

$$\mathcal{L}_{\text{FD}} = -\mathbb{E}_{v \in V} \left\| \varepsilon \left( \mathcal{R}_v(G + \Delta) - \varepsilon(I_v) \right) \right\|_2^2. \tag{10}$$

**Guidance Discrepancy Loss.** The guidance discrepancy loss is designed to ensure that the editing direction of the protected 3D scene deviates from that of the original scene under the same viewpoint. To achieve this, we utilize features from the noise space of the pre-trained InstructPix2Pix [2] model to represent guidance features. The loss is defined as follows:

$$\mathcal{L}_{\text{GD}} = -\mathbb{E}_{k,v,t,\epsilon,y_{\text{text}} \in Y_{\text{lib}}} \left\| \epsilon_\theta(I_v^k, \mathcal{R}_v(G + \Delta), y_{\text{text}}, t) - \epsilon_\theta(I_v', I_v, y_{\text{text}}, t) \right\|_2^2, \tag{11}$$

where $Y_{\text{lib}}$ represent a library of text prompt, as in DreamFusion [27], which contains 415 different prompts. It is used to introduce semantic diversity and prevent overfitting to a specific prompt.

**Overall Loss.** During training, the above two losses are jointly optimized with the main rendering loss, and the total loss can be expressed as:

$$\mathcal{L}_{\text{total}} = \mathcal{L}_{\text{render}} + \lambda_{\text{FD}} \cdot \mathcal{L}_{\text{FD}} + \lambda_{\text{GD}} \cdot \mathcal{L}_{\text{GD}}, \tag{12}$$

where $\mathcal{L}_{\text{render}}$ denotes the rendering loss, defined as the $L_2$ distance between rendered views and original views. $\lambda_{\text{FD}}$ and $\lambda_{\text{GD}}$ are the hyperparameters for the feature and guidance terms, respectively. The total loss is then used for view-focal gradient fusion to update the perturbation parameters.

## 3.4 View-Focal Gradient Fusion

During 3D object optimization, updates from different viewpoint may interfere with one another, leading to inconsistencies across multiple viewpoints. This effect is particularly noticeable in the optimization of fine-grained perturbations, resulting in extremely poor protection in some viewpoints. To mitigate this issue, we devise a View-Focal Gradient Fusion (VFGF) strategy that consists of two parts, a wide-view sampling and a view-focal weighting, which aims to improve the consistency and effectiveness of the perturbations across multiple views.

**Wide-View Sampling.** To ensure a uniform spatial distribution and broad coverage of the selected viewpoints, we design the wide-view sampling strategy to avoid concentration in locally similar or redundant views. For the a reference viewpoint $v_i$ and $v_j$, each is associated with a camera translation vector $\mathbf{T} \in \mathbb{R}^3$ and the rotation matrix $\mathbf{R} \in \text{SO}(3)$. We define the distance between two viewpoints as the sum of normalized translation and angular differences:

$$d(v_i, v_j) = \underbrace{\frac{\|\mathbf{T}_i - \mathbf{T}_j\|_2}{D_{\text{max}}}}_{\text{Translation distance}} + \underbrace{\frac{\angle(\mathbf{R}_i, \mathbf{R}_j)}{\pi}}_{\text{Rotation angle}}, \tag{13}$$

$$\angle(\mathbf{R}_i, \mathbf{R}_j) = \arccos\left( \frac{trace(\mathbf{R}_j \mathbf{R}_i^\top) - 1}{2} \right), \tag{14}$$

where $D_{\text{max}}$ is the maximum translation distance in view space and $trace(\cdot)$ denotes the trace of the matrix. Both components are mapped to $[0, 1]$ and equally weighted. Start from a initial set of viewpoints $V_1 = \{v_{\text{ref}}\}$, we select the next viewpoint furthest from all previously selected viewpoints based on all the statistics until $N$ viewpoints have been selected:

$$V_N = V_{N-1} \cup \left\{ v^* \mid v^* = \arg\max_{v_j \in \mathcal{V} \setminus V_{N-1}} \min_{v_i \in V_{N-1}} d(v_i, v_j) \right\}. \tag{15}$$

At each iteration, from the unselected viewpoints, find that viewpoint that has the largest distance from the nearest point in the current sampling set and add it to the sampling set. This sampling strategy avoids focusing the samples on locally similar viewpoints, thus providing stronger global structural constraints for the subsequent gradient fusion.

**View-Focal Weighting.** For the widely sampled set of viewpoints, we further propose a view-focal weighting strategy, which weights the gradient according to the difficulty of each viewpoint for

subsequent perturbation updates. Specifically, we treat the loss associated with each viewpoint as an indicator of its resistance to editing, and assign weights to the viewpoints based on the magnitude of the loss value, formally defined as follows:

$$\nabla_\Theta \mathcal{L}_{\text{focal}} = \sum_{v \in V} w_v \cdot \nabla_\Theta \mathcal{L}_{\text{total}}^v, \quad \text{where} \quad w_v = \frac{(\mathcal{L}_{\text{total}}^v + \tau)^\gamma}{\sum_{v' \in V} (\mathcal{L}_{\text{total}}^{v'} + \tau)^\gamma}, \tag{16}$$

where $\mathcal{L}_{\text{total}}^v$ denotes the total loss at viewpoint $v$, $\gamma$ is focusing parameter controlling the degree of nonlinearity in the weighting, and $\tau$ is a stabilization term to prevent zero values. This strategy biases the optimizer toward viewpoints with higher loss values (i.e., with weak defenses against editing), thereby generating perturbations that are effective across views to improve resistance to editing.

Benefiting from the above designs, our DEGauss effectively optimize 3D perturbations in space while maintaining perturbation validity across different views. This enables stable and view-generalizable protection against malicious 3D editing throughout the iterative optimization process.

## 4 Experiments

### 4.1 Experimental Setup

**Datasets and Editing Models.** We verified the effectiveness on common 3D editing dataset [11, 32, 1], including 'face', 'girl', 'person', 'bear', 'bicycle', and 'garden' scenes, with varying viewpoints and data scales. In addition, we validate our generalization leveraging the latest released 3D editing models, including GaussianEditor [7], DGE [4], DreamCatalyst [18], and EditSplat [19].

**Baselines.** Since the defense schemes for malicious 3D editing are still pending, we compare our DEGauss with state-of-the-art 2D editing defense schemes and migrated them to 3DGS to ensure fairness, including AdvDM [22], Mist [21], SDS [36], and AdvPaint [16].

**Evaluation Metrics.** We utilize Peak Signal-to-Noise Ratio (PSNR) to measure the difference between protected and original samples, reflecting the imperceptibility of perturbations. In addition, Contrastive Language-Image Pretraining (CLIP) similarity evaluates the semantic gap between protected and normal editing results. CLIP-T measures the semantic distance between the protected result and the editing prompt. CLIP Direction similarity (CLIP-D) [9] quantifies the consistency between text differences (from source to editing) and image differences (from source to edited).

**Implementation Details.** All experiments are conducted on a single NVIDIA RTX 4090 GPU. We set the number of sampled views $N = 6$, the weighting factor $\tau = 1e\text{-}6$, and $\gamma = 1.0$. To balance loss terms, we set hyperparameters $\lambda_{\text{FD}} = \lambda_{\text{GD}} = 1e\text{-}5$. The total number of training steps is set to 2,000. More settings and algorithm are provided in the **Supplementary**.

### 4.2 Comparisons

We compare existing 2D-based editing defense methods with our DEGauss, focusing on the ability to resist malicious editing in 3D scenarios. As shown in Fig. 3, existing 2D defense methods suffer from obvious editing artifacts and multi-view inconsistencies when directly applied to 3D scenes. For example, AdvDM [22] and Mist [21] introduce some noise or artifacts that degrade the quality of the edits, but still maintain normal editing effects without preserving the original scene detail. SDS [36] exhibit weaker suppression of malicious edits, offering limited resistance. Although AdvPaint [16] performs slightly better in suppressing editing effects, it still introduces noticeable editing traces, noise, and inconsistencies across different viewpoints. In contrast, our DEGauss better preserves the global structure and appearance of the original scene, producing natural and artifact-free renderings with high visual fidelity across multiple views.

Moreover, Table 1 quantitatively evaluates our DEGauss against existing methods. It can be seen that our DEGauss achieves the highest PSNR, indicating that the introduced perturbations are more imperceptible. Regarding CLIP-based scores, our DEGauss attains the lowest similarity between the edited and original samples, demonstrating its effectiveness in disrupting malicious editing trajectories. In contrast, existing 2D defense methods such as AdvDM and Mist exhibit significantly lower performance in all metrics. Even AdvPaint, which performs relatively better among the baselines, still lags behind DEGauss on several metrics. Both visual and quantitative comparisons

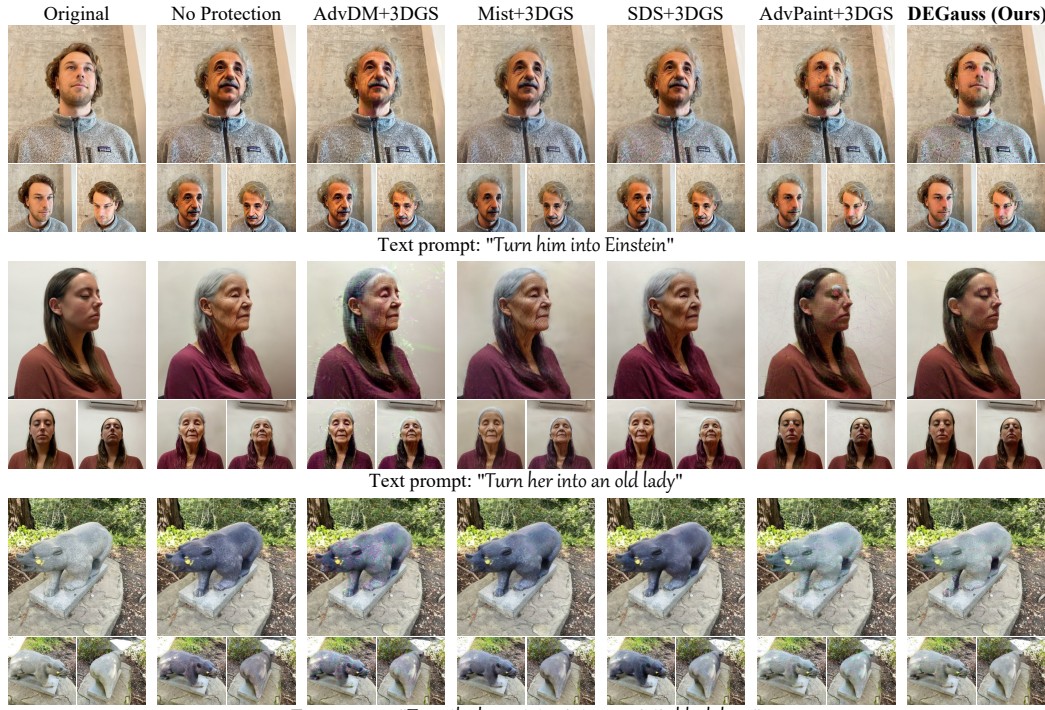

| Original | No Protection | AdvDM+3DGS | Mist+3DGS | SDS+3DGS | AdvPaint+3DGS | **DEGauss (Ours)** |

Text prompt: *"Turn him into Einstein"*

Text prompt: *"Turn her into an old lady"*

Text prompt: *"Turn the bear statue into an asiatic black bear"*

Figure 3: Visual comparison with existing methods. Comparisons include AdvDM [22], Mist [21], SDS [36], and AdvPaint [16]. Our DEGauss is resistant to 3D editing achieves optimal visualization in preserving the original 3D scene. The editing method used is GaussianEditor.

Table 1: Quantitative comparison with existing methods. Our DEGauss achieves the superior PSNR while offering the optimal resistance to 3D editing. The editing method used is GaussianEditor. Red indicates optimal and orange indicates suboptimal. ↑: higher is better, ↓: lower is better.

| Method | face | | | | girl | | | |
|---|---|---|---|---|---|---|---|---|
| | PSNR↑ | CLIP↓ | CLIP-T↓ | CLIP-D↓ | PSNR↑ | CLIP↓ | CLIP-T↓ | CLIP-D↓ |
| AdvDM [22]+3DGS | 32.63 | 0.9235 | 0.2465 | 0.0710 | 30.83 | 0.8716 | 0.2401 | 0.0586 |
| Mist [21]+3DGS | 33.63 | 0.9599 | 0.2539 | 0.0828 | 32.63 | 0.9131 | 0.2588 | 0.0872 |
| SDS [36]+3DGS | 32.39 | 0.9707 | 0.2493 | 0.0863 | 34.97 | 0.9573 | 0.2631 | 0.1071 |
| AdvPaint [16]+3DGS | 32.95 | 0.8996 | 0.2266 | 0.0355 | 32.93 | 0.8514 | 0.2434 | 0.0464 |
| **DEGauss (Ours)** | 33.92 | 0.8860 | 0.2193 | 0.0325 | 35.59 | 0.8689 | 0.2380 | 0.0446 |
| Method | bear | | | | bicycle | | | |
| | PSNR↑ | CLIP↓ | CLIP-T↓ | CLIP-D↓ | PSNR↑ | CLIP↓ | CLIP-T↓ | CLIP-D↓ |
| AdvDM [22]+3DGS | 29.26 | 0.9072 | 0.3057 | 0.0378 | 29.84 | 0.9385 | 0.2514 | 0.0732 |
| Mist [21]+3DGS | 31.10 | 0.9564 | 0.3106 | 0.0373 | 32.52 | 0.9675 | 0.2380 | 0.0567 |
| SDS [36]+3DGS | 31.30 | 0.9546 | 0.3107 | 0.0378 | 32.59 | 0.9724 | 0.2415 | 0.0616 |
| AdvPaint [16]+3DGS | 30.77 | 0.9046 | 0.3011 | 0.0314 | 30.89 | 0.9257 | 0.2464 | 0.0576 |
| **DEGauss (ours)** | 32.20 | 0.8949 | 0.2940 | 0.0256 | 32.84 | 0.9358 | 0.2336 | 0.0535 |

clearly illustrate that our DEGauss is able to effectively suppress malicious editing while maintaining a higher degree of visual consistency with the original 3D scene.

## 4.3 Generalization

To further evaluate the generalization of our DEGauss, we conduct experiments on SOTA different 3D editing methods, including GaussianEditor [7], DGE [4], DreamCatalyst [18] and EditSplat [19]. As shown in Fig. 4, our DEGauss preserves the original 3D appearance against GaussianEditor without introducing visible artifacts. In addition, for DGE and EditSplat, although subtle color modifications are observed, our DEGauss effectively prevents structural editing and maintains overall geometric

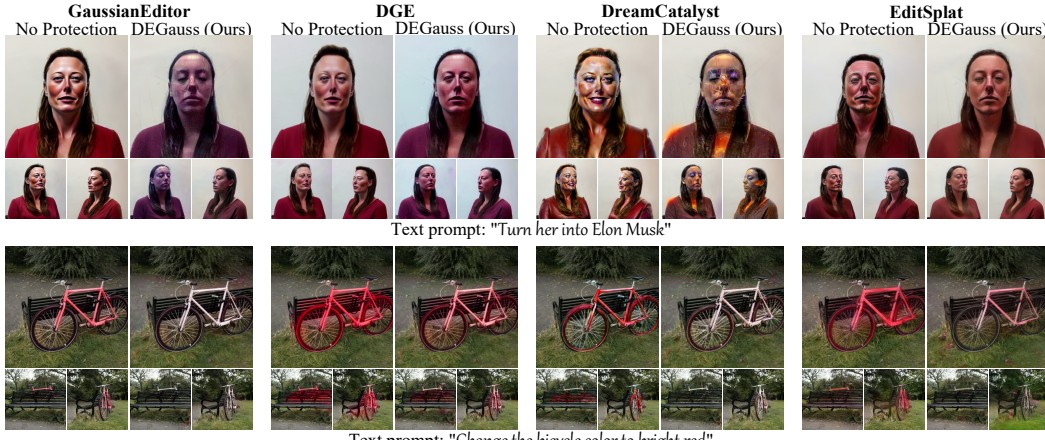

Figure 4: Visualization of generalization experiments. The editing methods include GaussianEditor [7], DGE [4], DreamCatalyst [18], and EditSplat [19]. Our DEGauss significantly resist 3D editing from different existing schemes.

Table 2: Quantitative analysis of generalization experiments. Our DEGauss is outstanding against different editing schemes. "Ori" and "Ours" refer to the non-protected and our protection results, and "Diff" refers to the difference between "Ori" and "Ours". ↑: higher is better, ↓: lower is better.

| | | face | | | | | | | |
| | CLIP | | | CLIP-T | | | CLIP-D | | |
| Method | Ori | Our↓ | Diff↑ | Ori | Our↓ | Diff↑ | Ori | Our↓ | Diff↑ |
|---|---|---|---|---|---|---|---|---|---|
| GaussianEditor [7] | 1.000 | 0.886 | 0.114 | 0.284 | 0.219 | 0.065 | 0.084 | 0.033 | 0.051 |
| DGE [4] | 1.000 | 0.890 | 0.110 | 0.253 | 0.230 | 0.023 | 0.089 | 0.050 | 0.039 |
| DreamCatalyst [18] | 1.000 | 0.896 | 0.104 | 0.277 | 0.248 | 0.029 | 0.089 | 0.044 | 0.045 |
| EditSplat [19] | 1.000 | 0.933 | 0.067 | 0.248 | 0.236 | 0.012 | 0.083 | 0.053 | 0.030 |
| | | girl | | | | | | | |
| | CLIP | | | CLIP-T | | | CLIP-D | | |
| Method | Ori | Our↓ | Diff↑ | Ori | Our↓ | Diff↑ | Ori | Our↓ | Diff↑ |
| GaussianEditor [7] | 1.000 | 0.869 | 0.131 | 0.260 | 0.238 | 0.022 | 0.120 | 0.045 | 0.075 |
| DGE [4] | 1.000 | 0.919 | 0.081 | 0.259 | 0.250 | 0.009 | 0.119 | 0.067 | 0.052 |
| DreamCatalyst [18] | 1.000 | 0.821 | 0.179 | 0.265 | 0.244 | 0.021 | 0.096 | 0.064 | 0.032 |
| EditSplat [19] | 1.000 | 0.942 | 0.058 | 0.276 | 0.265 | 0.011 | 0.124 | 0.085 | 0.039 |
| | | bear | | | | | | | |
| | CLIP | | | CLIP-T | | | CLIP-D | | |
| Method | Ori | Our↓ | Diff↑ | Ori | Our↓ | Diff↑ | Ori | Our↓ | Diff↑ |
| GaussianEditor [7] | 1.000 | 0.895 | 0.105 | 0.313 | 0.294 | 0.019 | 0.062 | 0.026 | 0.036 |
| DGE [4] | 1.000 | 0.890 | 0.110 | 0.310 | 0.298 | 0.012 | 0.071 | 0.023 | 0.048 |
| DreamCatalyst [18] | 1.000 | 0.917 | 0.083 | 0.333 | 0.305 | 0.028 | 0.074 | 0.035 | 0.039 |
| EditSplat [19] | 1.000 | 0.935 | 0.065 | 0.312 | 0.309 | 0.003 | 0.051 | 0.028 | 0.023 |
| | | bicycle | | | | | | | |
| | CLIP | | | CLIP-T | | | CLIP-D | | |
| Method | Ori | Our↓ | Diff↑ | Ori | Our↓ | Diff↑ | Ori | Our↓ | Diff↑ |
| GaussianEditor [7] | 1.000 | 0.939 | 0.061 | 0.270 | 0.234 | 0.036 | 0.076 | 0.054 | 0.022 |
| DGE [4] | 1.000 | 0.956 | 0.044 | 0.293 | 0.261 | 0.032 | 0.098 | 0.088 | 0.010 |
| DreamCatalyst [18] | 1.000 | 0.955 | 0.045 | 0.269 | 0.243 | 0.026 | 0.092 | 0.057 | 0.035 |
| EditSplat [19] | 1.000 | 0.951 | 0.049 | 0.280 | 0.272 | 0.008 | 0.116 | 0.093 | 0.023 |

and semantic integrity. As for DreamCatalyst, our perturbations lead to the generation of noisy and chaotic outputs, significantly disrupting the intended editing direction. These results collectively demonstrate that our DEGauss robustly resists unauthorized 3D editing in the state-of-the-art editing pipelines, highlighting its strong generalization to diverse editing paradigms.

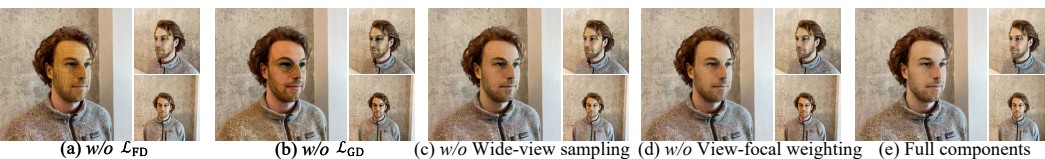

(a) *w/o* $\mathcal{L}_{FD}$     (b) *w/o* $\mathcal{L}_{GD}$     (c) *w/o* Wide-view sampling     (d) *w/o* View-focal weighting     (e) Full components

Figure 5: Visualization of the ablation on key components. The editing model is GaussianEditor and the text prompt is "Make him wear a Venetian mask".

Table 2 reports the CLIP-based similarity scores between our DEGauss results and the original samples under different 3D editing methods. The samples protected by DEGauss consistently achieve significantly lower similarity under all evaluated editors, indicating that the introduced perturbations effectively disrupt the expected editing trajectories regardless of the editing scheme applied.

## 4.4 Ablation Studies

**Ablation on Key Components.** We perform an ablation study on the key components of our framework, the visual results are shown Fig. 5. When the feature discrepancy loss is removed, the edited images exhibit noticeable degradation in quality. While without the guidance discrepancy loss, the outputs retain the intended semantic edits, leading to the weakest defense performance. In the absence of wide-view sampling or view-focus weighting, the results inconsistent in multi-view, i.e., some views are effectively defended while others are poorly. In contrast, when all components are integrated, our method achieves the strongest overall protection against malicious editing.

**Ablation on Hyperparameters.** We verify the effect of noise strength on stealthiness by uniformly adjusting the perturbation hyperparameter $\lambda$ (jointly $\lambda_{FD}$ and $\lambda_{GD}$), as shown in Table 3. It can be observed that reducing $\lambda$ improves noise stealthiness (higher PSNR score), while decreasing defense performance (higher CLIP scores). When $\lambda = 1e$-5, noise invisibility and defense ability reach a trade-off.

Table 3: Noise Stealthiness vs. Defense Strength. Red indicates optimal and orange indicates suboptimal.

|       | PSNR↑ | CLIP↓  | CLIP-T↓ | CLIP-D↓ |
|-------|-------|--------|---------|---------|
| $1e$-4 | 27.47 | 0.8147 | 0.2018  | 0.0171  |
| $1e$-5 | 33.92 | 0.8860 | 0.2193  | 0.0325  |
| $1e$-6 | 34.80 | 0.9617 | 0.2692  | 0.0555  |

We conduct an ablation study on the hyperparameters of loss function, as shown in Table 4. When the weight of the feature discrepancy loss is reduced, the PSNR score drops to lowest, indicating poor visual quality in the generated samples. Alternatively, decreasing the weight of the guidance discrepancy loss leads to the highest PSNR, but the defense becomes ineffective and the edited result will retain more of the expected semantics. In contrast, our chose hyperparameter setting achieves a balanced trade-off between visual fidelity and editing resistance.

Table 4: Ablation experiments on hyperparameters. The editing model is GaussianEditor. Red indicates optimal and orange indicates suboptimal.

| $\lambda_{FD}$ | $\lambda_{GD}$ | PSNR↑ | CLIP↓  | CLIP-T↓ | CLIP-D↓ |
|---------|---------|-------|--------|---------|---------|
| *0.1    | -       | 31.56 | 0.8825 | 0.2297  | 0.0594  |
| -       | *0.1    | 36.17 | 0.9244 | 0.2295  | 0.1020  |
| -       | -       | 33.92 | 0.8860 | 0.2193  | 0.0325  |

## 5 Conclusions

In this paper, we propose DEGauss, a novel defense framework dedicated to protecting 3DGS from malicious editing. Unlike 2D defense approaches that focus only on image space, our DEGauss optimizes perturbations in 3D space to maintain consistent protection across multiple views. Specifically, we elaborate the view-focal gradient fusion mechanism and the dual-discrepancy optimization strategy that jointly disturb the direction of multi-view editing during iterative editing, thus preventing the expected semantic editing. Experimental results indicate that our DEGauss can effectively defend unauthorized 3D editing in different scenes while maintaining high fidelity visual quality.

## Acknowledgments

The authors are very indebted to the anonymous referees for their critical comments and suggestions for the improvement of this paper. This work was supported by the National Key Research and Development Program of China (2021YFA1000102), National Natural Science Foundation of China (Nos. 62376285, 61673396), Natural Science Foundation of Shandong Province (No: ZR2022MF260).

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
