# OpenReview forum: "DEGauss: Defending Against Malicious 3D Editing for Gaussian Splatting"
_NeurIPS.cc/2025/Conference — NeurIPS 2025 poster_

### Official Review · Reviewer_2tt7 · 2025-06-20

**Clarity:** 3
**Significance:** 3
**Originality:** 3
**Rating:** 4
**Confidence:** 4

**Summary:**

- This paper presents DEGauss, a novel defense framework designed to resist malicious 3D editing in Gaussian splatting scenarios.

- The authors address the limitations of existing 2D defense methods, which fail to handle 3D spatial correlations and are prone to perturbation elimination during 3D editing iterations.

**Questions:**

- How did the author optimize Equation 12? The Algorithm 1 in the Supplementary Material is not clear.
- The comparison results between DEGauss and baselines are missing in Table 2.
- Why doesn't the author output distorted content after malicious editing like in [1]?


[1] Ruiz, N., Bargal, S. A., & Sclaroff, S. (2020). Disrupting deepfakes: Adversarial attacks against conditional image translation networks and facial manipulation systems. In Computer Vision–ECCV 2020 Workshops: Glasgow, UK, August 23–28, 2020, Proceedings, Part IV 16 (pp. 236-251). Springer International Publishing.

**Ethical Concerns:**

["NO or VERY MINOR ethics concerns only"]

**Final Justification:**

- The authors have addressed my main concern.

**Limitations:**

- Although this paper represents a significant piece of work, there are several aspects that are difficult to comprehend (please see Questions for details), and the writing quality of the paper needs to be improved.

**Paper Formatting Concerns:**

- There are no obvious format concerns in this paper.

**Quality:**

3

**Strengths And Weaknesses:**

**Strengths:**
- DEGauss is the first framework specifically tailored for defending against 3D malicious editing.

**Weaknesses:**
- DEGauss is designed for Gaussian splatting and may not directly translate to other 3D formats (e.g., meshes, NeRFs) without further adaptation.
- The wide-view sampling and iterative optimization might introduce higher computational costs compared to 2D defense methods, though the paper does not explicitly quantify this.
- The framework of DEGauss relies on pre-trained diffusion models (e.g., InstructPix2Pix), which could limit its performance if these models have inherent biases.

---

> ### Author Rebuttal · Authors · 2025-07-30
>
> Thank you for your valuable comments and for your contribution to improving our paper. We have carefully considered your feedback and provide detailed, point-by-point responses as follows:
>
> **Weakness 1: Explanation of transferring DEGauss to other 3D formats (e.g. meshes, NeRFs).**
>
> **Answer:** Thank you for your constructive feedback. While our perturbation modeling is tailored to 3D Gaussian splatting, our main strategies View-Focal Gradient Fusion and Dual Discrepancy Optimization are not restricted to this representation and can be readily adapted to other 3D formats without further adaptation. For example, in meshes or NeRF, when optimizing multiple viewpoints, our View-Focal Gradient Fusion strategy can also be used to apply weights to different viewpoints. These strategies are equally important for other 3D representations, as ensuring multi-view consistency and adaptive gradient emphasis are fundamental challenges across different 3D domains. We will further explore such generalizations in future work.
>
> **Weakness 2: Comparison of computational costs with 2D defense methods.**
>
> **Answer:** Indeed, due to the multi-view nature of 3D optimization, there is an inherent disadvantage in terms of computational cost compared to 2D settings. To ensure fairness, we compared the time consumption and GPU memory usage of the baseline methods extended to 3D defense under the same settings. As shown in Table R1, the wall-clock time and GPU memory of our method is slightly higher than that of AdvDM, Mist, and SDS, due to the additional computational overhead introduced by the multi-view weighting strategy. However, our cost is shorter than AdvPaint, as we only compute gradients using a subset of feature maps rather than the full attention map used in AdvPaint.
>
> It is important to note that, although some 2D-based baseline methods may achieve shorter time in simpler settings, they fail to meet the multi-view consistency requirements necessary for effective 3D protection (higher CLIP scores). In contrast, our DEGauss framework achieves superior protection across all views, while maintaining comparable computational cost to other methods. We will include the complete table and related discussion in the revised version.
>
> Table R1:  Comparison of computational costs and attack performance with 2D defense methods. Bold indicates the best result.
>
> |  | AdvDM+3DGS | Mist+3DGS | SDS+3DGS | AdvPaint+3DGS | DEGauss (Ours) |
> |:---:|:---:|:---:|:---:|:---:|:---:|
> | Wall-Clock Time $\downarrow$ | 520.06s | 675.49s | **361.01s** | 959.08s | 893.07s |
> | GPU Memory $\downarrow$ | 15112MiB | 16744MiB | **13508MiB** | 20220MiB | 16938MiB |
> | CLIP $\downarrow$ | 0.9235 | 0.9599 | 0.9707 | 0.8996 | **0.8860** |
>
> **Weakness 3: Reliance on pre-trained diffusion models.**
>
> **Answer:** Thank you for your valuable feedback.
> We would like to clarify that the most advanced methods [1,2,3,4,5] for defending against malicious editing currently rely on pre-trained diffusion models, which are used to optimize perturbations through gradient backpropagation, thereby disrupting unauthorized editing in homogenous diffusion models. We follow this paradigm in our approach and demonstrate superior performance, while also exhibiting excellent generalizability to other mainstream editing frameworks, as shown in Table 2 of main paper. Despite the fact that reliance on pre-trained diffusion models inevitably introduces inherent biases, it remains a widely accepted paradigm in this field. We will point out this limitation in the revised paper.
>
> [1] Adversarial Example Does Good: Preventing Painting Imitation from Diffusion Models via Adversarial Examples. ICML, 2023.
>
> [2] Raising the Cost of Malicious AI-Powered Image Editing. ICML, 2023.
>
> [3] EditShield: Protecting Unauthorized Image Editing by Instruction-guided Diffusion Models. ECCV, 2024.
>
> [4] Edit Away and My Face Will Not Stay: Personal Biometric Defense against Malicious Generative Editing, CVPR, 2025.
>
> [5] DiffusionGuard: A Robust Defense Against Malicious Diffusion-based Image Editing. ICLR, 2025.
>
>
> **Question 1: How did the author optimize Equation 12?**
>
> **Answer:** Thank you for your comment. The optimization of Equation 12 follows these steps:
>
> a). Overall Loss Calculation: For each view $i$, we compute the total loss $L_{total}^{i}$.
>
> b). Focal Weight Calculation: We then calculate the focal weights $w_i = \frac{(L_{\text{total}}^i + \epsilon)^\gamma}{\sum_{V_n} (L_{\text{total}} + \epsilon)^\gamma}$ for each view based on its loss, so that views with higher loss are given greater emphasis in subsequent gradient updates.
>
> c). Perturbation Update: The perturbation $\Delta^{(k)}$ of $k$ iteration is updated by aggregating the weighted gradients $\sum_{n=1}^{N} w_n \cdot \nabla_{\Theta^{(k-1)}} L_{\text{total}}^{n}$ from all sampled views, following a PGD-style step $\Delta^{(k)} = \Delta^{(k-1)} - \eta \cdot \text{sign}(\sum_{n=1}^{N} w_n \cdot \nabla_{\Theta^{(k-1)}} L_{\text{total}}^{n})$, where $\text{sign}(.)$ represents a symbolic function, which takes the value 1 or -1 depending on whether the value is positive or negative.
>
> d). Parameter Update: Finally, the parameters of Gaussians $\Theta^{(k)}$ are optimized by adding the perturbation $\Theta^{(k)} = \Theta^{(k-1)}+\Delta^{(k)}$.
>
> We will provide a detailed explanation of Algorithm 1 in the supplementary materials, along with step-by-step pseudocode to ensure clarity and ease of understanding.
>
> **Question 2: Baselines are missing in Table 2.**
>
> **Answer:** Thank you for your suggestion. We have added the baseline (non-protected) results and the differences before and after protection to the Table R2 to provide a clearer comparison of attack effectiveness. It is observed that our DEGauss significantly and consistently reduces the CLIP similarity with the normal editing results, more intuitively demonstrating the effectiveness of our method under different editing schemes. We only present results for two scenes due to page limitations. The updated complete table and related discussion will be added in the revised version.
>
> Table R2: Quantitative analysis of generalization. “Ori” and “Ours” refer to the non-protected and our protection results, and “Diff” refers to the difference between “Ori” and “Ours”.
>
> |  |  |  |  |  |  | face |  |  |  |  |  |
> |---|:---:|:---:|:---:|---|:---:|:---:|:---:|---|:---:|:---:|:---:|
> |  |  | CLIP |  |  |  | CLIP-T |  |  |  | CLIP-D |  |
> | Method | Ori | Our $\downarrow$ | Diff $\uparrow$ |  | Ori | Our $\downarrow$ | Diff $\uparrow$ |  | Ori | Our $\downarrow$ | Diff $\uparrow$ |
> | GaussianEditor | 1.000  | 0.886  | 0.114  |  | 0.284  | 0.219  | 0.065  |  | 0.084  | 0.033  | 0.051  |
> | DGE | 1.000  | 0.890  | 0.110  |  | 0.253  | 0.230  | 0.023  |  | 0.089  | 0.050  | 0.039  |
> | DreamCatalyst | 1.000  | 0.896  | 0.104  |  | 0.277  | 0.248  | 0.029  |  | 0.089  | 0.044  | 0.045  |
> | EditSplat | 1.000  | 0.933  | 0.067  |  | 0.248  | 0.236  | 0.012  |  | 0.083  | 0.053  | 0.030  |
> |  |  |  |  |  |  | **bear** |  |  |  |  |  |
> |  |  | CLIP |  |  |  | CLIP-T |  |  |  | CLIP-D |  |
> | Method | Ori | Our $\downarrow$ | Diff $\uparrow$ |  | Ori | Our $\downarrow$ | Diff $\uparrow$ |  | Ori | Our $\downarrow$ | Diff $\uparrow$ |
> | GaussianEditor | 1.000  | 0.895  | 0.105  |  | 0.313  | 0.294  | 0.019  |  | 0.062  | 0.026  | 0.036  |
> | DGE | 1.000  | 0.890  | 0.110  |  | 0.310  | 0.298  | 0.012  |  | 0.071  | 0.023  | 0.048  |
> | DreamCatalyst | 1.000  | 0.917  | 0.083  |  | 0.333  | 0.305  | 0.028  |  | 0.074  | 0.035  | 0.039  |
> | EditSplat | 1.000  | 0.935  | 0.065  |  | 0.312  | 0.309  | 0.003  |  | 0.051  | 0.028  | 0.023  |
>
> **Question 3: Why not output distorted content after malicious editing?**
>
> **Answer:** Thank you for your insightful comment.
> Our primary goal is to protect personal 3D assets from unauthorized or malicious editing. There are two different defense paradigms in this task: (1) disrupting the output results of editing and (2) maintaining the original input appearance. The fomer, exemplified by method [1], disrupts editing by intentionally outputting highly distorted results, thereby degrading the edited asset. In contrast, the latter, which we follow, focuses on maintaining the similarity to the original input and suppressing the effectiveness of malicious editing without introducing severe deformation. This is particularly important for personal digital 3D facial avatars, as facial distortions can lead to the stigmatization or reputational harm of the content owner. Therefore, in scenarios where preserving digital identity integrity is essential, methods that deliberately produce distorted results are not appropriate.
>
> Accordingly, our strategy is to minimize any perceptible damage to the underlying 3D shape and appearance, ensuring that the original quality of the 3D asset is fully preserved. This user-centric defense paradigm is more consistent with real-world needs, enabling individuals to protect their 3D assets without sacrificing visual fidelity.
>
> [1] Ruiz, N., et al. Disrupting deepfakes: Adversarial attacks against conditional image translation networks and facial manipulation systems. ECCV, 2020.
>
> **Thank you again for your valuable feedback. We will revise the paper to incorporate all related experiments and detailed discussions as you suggested. If there are any additional issues, we are happy to provide further explanations.**

---

> ### Author Response · Authors · 2025-08-08
> **We sincerely hope response from the reviewer before the end of the author-reviewer discussion period.**
>
> Dear Reviewer 2tt7,
>
> Thank you for the careful evaluation of our paper. We want to know whether our previous response can relieve or successfully address any of your initial concerns. If not and you have any further questions, please feel free to share with us so that we can use our last chance to address them. We want to thank you again for your valuable time and efforts. We will be more than happy to receive your response and resolve any further problems.

---

### Official Review · Reviewer_2LZe · 2025-07-02

**Clarity:** 2
**Significance:** 3
**Originality:** 3
**Rating:** 5
**Confidence:** 4

**Summary:**

This paper proposes DEGauss, a novel adversarial attack framework targeting diffusion-based 3D Gaussian Splatting (3DGS) editing. Building upon AdvDM, a 2D adversarial attack formulation for image editing, the authors extend the approach to the 3DGS domain. To enhance attack effectiveness, they introduce two key loss components: Feature Discrepancy Loss and Guidance Discrepancy Loss, which help disrupt the editing process more robustly. Additionally, the paper presents View-Focal Gradient Fusion, a technique designed to ensure balanced viewpoint sampling and adaptive loss weighting in order to the optimization process of adversarial perturbations. As the first work to explore adversarial attacks in 3DGS editing, the paper conducts extensive experiments across various combinations of 2D editing and attack methods, demonstrating superior performance in disrupting the editing pipeline.

**Questions:**

1. Could you revise the notational conventions used throughout the paper, or provide an explicit explanation of how the semicolon is being used to distinguish function arguments? This would greatly improve clarity.
2. Are my interpretations of Equations 1, 3, 6, 8–9, and 11 correct? Specifically:
- Eq. 1: $D_\phi(I_v^k, I_v, y_{\text{text}})$
- Eq. 3: $\epsilon_\theta(x_t + \delta, y_{\text{text}}, t)$
- Eq. 6: $D_\phi(R_v(G^k + \Delta), y_{\text{text}})$
- Eq. 8, 9 and 11: $\epsilon(R_v(G^k + \Delta), y_{\text{text}}, t)$
3. Can you provide visualizations to support the claim in Line 107 that naïve 2D adversarial attacks fail when applied to 3DGS editing? This would strengthen the motivation for your method.
4. What are the relative contributions of each loss term (e.g., $L_{\text{FD}}$, $L_{\text{GD}}$, and $L_{\text{render}}$) to the final perturbation $\Delta$?
5. How exactly are the baseline methods adapted to the 3DGS setting? Do they follow the same procedure in Algorithm 1, with the difference only in the loss?
6. Could you please include the baseline similarity values (without DEGauss) in Table 2 to make the effect of your attack more transparent?

**Ethical Concerns:**

["NO or VERY MINOR ethics concerns only"]

**Final Justification:**

This paper is the first attempt to investigate the adversarial attack problem in the domain of 3DGS editing. The paper is technically solid and novel. Although there are some issues with the presentation, such as the equation and missing baseline results, the author's rebuttal successfully addresses the most of my concerns. Therefore, I would like to raise my rating to "accept".

**Limitations:**

Yes

**Paper Formatting Concerns:**

No formatting issue

**Quality:**

3

**Strengths And Weaknesses:**

**Strengths**
1. This paper introduces a novel and timely contribution by generalizing adversarial attacks from 2D image editing to 3D Gaussian Splatting (3DGS) editing. To the best of my knowledge, this is the first work to explore adversarial robustness in this context, marking a meaningful step forward in adversarial machine learning for 3D generative models. The mathematical formulation is sound and presented in a coherent manner.
2. The experimental evaluation is thorough and well-structured. The authors benchmark DEGauss against a wide range of combinations of 2D image editors and adversarial attack methods, demonstrating the effectiveness of the proposed approach in disrupting the editing process. Furthermore, extensive visualizations are provided, supporting the quantitative claims and enhancing the paper’s clarity.

**Weaknesses**
1. The paper frequently uses a non-standard notation format, e.g., writing functions as $f(a; b, c)$ instead of the more conventional $f(a, b, c)$. This style appears throughout key equations such as Eq. 1, Eq. 3, Eq. 6, and others, and introduces significant ambiguity, especially since the use of a semicolon does not follow established conventions (e.g., semicolon usually used in parameterized distributions like $\mathcal{N}(x; \mu, \sigma^2)$). The lack of an explanation for this notation makes it difficult to interpret equations precisely.
2. Due to the notational issues, several equations are difficult to parse. For instance, in Eq. 1, the editing model (e.g., InstructNeRF2NeRF) takes both the rendered image and the original image as input, so a clearer version might be written as $D_\phi(I_v^k, I_v, y_{\text{text}})$. Similarly, in Eq. 3, the denoising network should arguably be expressed as $\epsilon_\theta(x_t + \delta, y_{\text{text}}, t)$, and Eq. 6 should likely read $D_\phi(R_v(G^k + \Delta), y_{\text{text}})$. These should be revised to match standard notation and improve interoperability.
3. In the paragraph starting at Line 107, the authors discuss why 2D adversarial attacks fail when applied directly to the 3DGS setting. However, these arguments are entirely textual. Supporting visualizations would greatly aid readers in understanding the challenges and motivate the proposed modifications.
4. The rendering loss $L_{\text{render}}$ in Eq. 12 appears to play a critical role in constraining the adversarial perturbation $\Delta$, ensuring that the edited result remains close to the original rendering. However, this is only implied, not clearly stated. This term should be explicitly discussed as a regularizer, especially since the adversarial loss components $L_{\text{FD}}$ and $L_{\text{GD}}$ are given small weights (1e-5). It would be helpful to know how much each loss component contributes to the final perturbation in practice.
5. In the supplementary material (Line 452), the authors mention that they "converted the original optimizations and updates for 2D perturbation to updates for 3DGS parameters." However, it is unclear how the baselines are adapted to the 3DGS setting. Do they follow the same update pipeline as DEGauss in Algorithm 1, with only the loss function swapped? Clarifying this setup is important for understanding the fairness and consistency of the comparisons.
6. Table 2 is meant to demonstrate that DEGauss significantly reduces similarity across editors, indicating successful attack performance. However, it does not include baseline values (i.e., similarity scores without applying DEGauss), which makes the relative effectiveness of the attack hard to evaluate. Including these values would make the table much more informative and complete.

**Conclusion**

Despite some weaknesses in presentation and clarity, particularly around notation, equation interpretation, and baseline details, the core contribution of this paper is novel and promising. As the first attempt at designing adversarial attacks for 3DGS editing, it opens up a new research direction and provides a solid foundation for future work. Therefore, I currently lean toward borderline acceptance, and I would be inclined to raise my score if the authors can adequately address the above questions and concerns.

---

> ### Author Rebuttal · Authors · 2025-07-30
>
> Thank you for your constructive comments and for contributing to the improvement of our paper. We have carefully consider and address them point by point to your comments in the following responses.
>
> **Weakness 1 & Question 1: Explanation of semicolon in Equations.**
>
> **Answer:** Thanks for your valuable suggestion. As suggested, we have replaced all semicolons with commas in all equations in the paper to avoid ambiguity and ensure that the formulas are easy to understand. Our original intention in using semicolons (“;”) was to distinguish between main inputs and conditional inputs in the equations. However, we agree that this notation may lead to confusion or misinterpretation. To improve readability and maintain consistency with common academic conventions, we have now unified the notation and used commas throughout the paper. We appreciate your careful review and constructive feedback on this detail.
>
> **Weakness 2 & Question 2: Explanation of the Equations for editing and denoising models.**
>
> **Answer:** Yes, your understanding is generally correct. We initially used semicolons to separate the main input and conditional input, but this seems to be misleading. In fact, all symbols in the Equations are inputs. Therefore, in Equations (1) and (6), the rendered image $I_v^k$ also needs to be used as the main input for the editing process to perform the editing operations of the diffusion model. Thus, they should be modified as follows:
>
> Eq. (1): $D_{\phi}(I_v^k, I_v, y_{\text{text}})$,
> Eq. (6): $D_{\phi}(I_v^k, R_v(G^k + \Delta), y_{\text{text}})$.
>
> For the denoising network, which is a key component of the diffusion model, the above inputs are also continued, so Equations (8), (9), and (11) should be rewritten as follows:
>
> $\epsilon_{\theta}(I_v^k, R_v(G^k + \Delta), y_{\text{text}}, t)$.
>
> Similarly, in Equation (3), the noisy image $x_t$ after the $t$-th noise addition to the original image also needs to be used as input, as follows:
>
> $\epsilon_{\theta}(x_t, x + \delta, y_{\text{text}}, t)$.
>
> These modifications ensure that all relevant variables are clearly and consistently represented in the equations.
>
> **Weakness 3 & Question 3: Can you provide visualizations to support that naïve 2D adversarial attacks fail when applied to 3DGS editing?**
>
> **Answer:** Thanks for your constructive suggestion. We have observed several clear cases where simple 2D adversarial attacks fail when applied to 3DGS setting, as shown in Figure 3 of main paper. However, due to the rebuttal format restrictions, we are unable to provide images, PDFs, or external links on this page. We will add a more intuitive visual demonstration to reinforce this motivation into the main paper, which will further help readers understand these challenges more clearly.
> Specifically, the core reason why 2D adversarial attacks fail in 3DGS is the lack of multi-view consistency. Perturbations designed for a single view struggle to remain effective in all viewpoints. Furthermore, 2D attacks operate on pixel space and lack control over 3D geometric structure, thus they are ineffective in 3D space.
>
> **Weakness 4 & Question 4: Contributions of each loss component.**
>
> **Answer:** Thank you for your valuable comment. As shown in Table 3 of main paper, we have conducted an ablation study on the weights of $L_{FD}$ and $L_{GD}$, and found that setting both to $1e\mbox{-}5$ yields optimal performance. These weights are set to a small value because their magnitude differs significantly from that of $L_{render}$.
>
> Additionally, as suggested, we have performed a comprehensive ablation study to quantify the impact of the attack losses ($L_{FD}$ and $L_{GD}$) and the regularization term $L_{render}$. Specifically, we fix the weight of the rendering loss to 1 as a control variable and adjust the weights of $L_{FD}$ and $L_{GD}$ (jointly denoted as $\lambda$, as these two are set in balance) to evaluate their influence. As show in the Table R1, increasing the weight $\lambda$ enhances attack strength (lower CLIP scores) but reduces stealthiness (higher PSNR scores), while decreasing their weight $\lambda$ improves stealth but weakens the attack effect. When the weight $\lambda =1e\mbox{-}5$, the best balance between protection strength and visual fidelity can be achieved. This indicates that attack losses $L_{FD}$ and $L_{GD}$ make important contributions to edit resistance, while rendering loss $L_{render}$ plays a key regularizing role in maintaining the visual quality and structural integrity of 3D assets.
>
> Table R1: Ablation study of each loss component. **Bold** indicates the best result.
>
> | $\lambda$ | PSNR $\uparrow$ | CLIP $\downarrow$ | CLIP-T $\downarrow$ | CLIP-D $\downarrow$ |
> |---|:---:|:---:|:---:|:---:|
> | $1e\mbox{-}4$ | 27.47  | **0.8147** | **0.2018** | **0.0171** |
> | $1e\mbox{-}5$ | 33.92  | 0.8860  | 0.2193  | 0.0325  |
> | $1e\mbox{-}6$ | **34.80** | 0.9617  | 0.2692  | 0.0555  |
>
> **Weakness 5 & Question 5: How exactly are the baseline methods adapted to the 3DGS setting?**
>
> **Answer:** Yes, all baseline methods follow the same process as Algorithm 1 and utilize their unique losses to achieve perturbation updates for 3DGS. This is to ensure fairness in comparison. We will provide detailed descriptions in the revised version to ensure that all baselines are described clearly and completely.
>
> **Weakness 6 & Question 6: Include the baseline similarity values (without DEGauss) in Table 2.**
>
> **Answer:** Thank you for your suggestion. We have added the baseline (non-protected) results and the differences before and after protection to the Table R2 to provide a clearer comparison of attack effectiveness. It can be seen that our DEGauss significantly and consistently reduces the CLIP similarity with the normal editing results, more intuitively demonstrating the effectiveness of our method under different editing schemes. We only present results for two scenes due to page limitations. The updated complete table and related discussion will be added in the revised version.
>
> Table R2:  Quantitative analysis of generalization. “Ori” and “Ours” refer to the non-protected and our protection results, and “Diff” refers to the difference between “Ori” and “Ours”.
>
> |  |  |  |  |  |  | face |  |  |  |  |  |
> |---|:---:|:---:|:---:|---|:---:|:---:|:---:|---|:---:|:---:|:---:|
> |  |  | CLIP |  |  |  | CLIP-T |  |  |  | CLIP-D |  |
> | Method | Ori | Our $\downarrow$ | Diff $\uparrow$ |  | Ori | Our $\downarrow$ | Diff $\uparrow$ |  | Ori | Our $\downarrow$ | Diff $\uparrow$ |
> | GaussianEditor | 1.000  | 0.886  | 0.114  |  | 0.284  | 0.219  | 0.065  |  | 0.084  | 0.033  | 0.051  |
> | DGE | 1.000  | 0.890  | 0.110  |  | 0.253  | 0.230  | 0.023  |  | 0.089  | 0.050  | 0.039  |
> | DreamCatalyst | 1.000  | 0.896  | 0.104  |  | 0.277  | 0.248  | 0.029  |  | 0.089  | 0.044  | 0.045  |
> | EditSplat | 1.000  | 0.933  | 0.067  |  | 0.248  | 0.236  | 0.012  |  | 0.083  | 0.053  | 0.030  |
> |  |  |  |  |  |  | **bear** |  |  |  |  |  |
> |  |  | CLIP |  |  |  | CLIP-T |  |  |  | CLIP-D |  |
> | Method | Ori | Our $\downarrow$ | Diff $\uparrow$ |  | Ori | Our $\downarrow$ | Diff $\uparrow$ |  | Ori | Our $\downarrow$ | Diff $\uparrow$ |
> | GaussianEditor | 1.000  | 0.895  | 0.105  |  | 0.313  | 0.294  | 0.019  |  | 0.062  | 0.026  | 0.036  |
> | DGE | 1.000  | 0.890  | 0.110  |  | 0.310  | 0.298  | 0.012  |  | 0.071  | 0.023  | 0.048  |
> | DreamCatalyst | 1.000  | 0.917  | 0.083  |  | 0.333  | 0.305  | 0.028  |  | 0.074  | 0.035  | 0.039  |
> | EditSplat | 1.000  | 0.935  | 0.065  |  | 0.312  | 0.309  | 0.003  |  | 0.051  | 0.028  | 0.023  |
>
> **Thanks again for your constructive comments. We will address your comments by including all pertinent experiments and supplementary details in our revision. If you need further explanations, we are happy to elaborate in more detail.**

---

> > ### Comment · Reviewer_2LZe · 2025-08-01
> >
> > I thank the authors for their detailed response, which solve most of my concerns, especially my confusion on the equation and the contribution of each loss term. Therefore, I would like to raise my score to "accept".

---

> > > ### Author Response · Authors · 2025-08-01
> > >
> > > Thank you for taking the time to review our paper and for providing your insightful comments. We sincerely appreciate your thoughtful evaluation and are pleased to hear that our rebuttal addressed most of your concerns.

---

### Official Review · Reviewer_Uu15 · 2025-07-02

**Clarity:** 2
**Significance:** 2
**Originality:** 2
**Rating:** 3
**Confidence:** 5

**Summary:**

The authors propose DEGauss, a novel defense framework targeting malicious 3D editing in Gaussian Splatting. Unlike prior 2D defenses, DEGauss targets on building spatial correlations across multiple views and addresses the limitations of iterative degradation. The method introduces a view-focal gradient fusion mechanism to adaptively prioritize perturbations across viewpoints, and a dual discrepancy optimization strategy to jointly maximize semantic and directional editing divergence. Experiments across diverse scenes and state-of-the-art editing show that DEGauss provides robust multi-view resistance.

**Questions:**

See the weaknesses

**Ethical Concerns:**

["NO or VERY MINOR ethics concerns only"]

**Final Justification:**

I am convinced with the W1 and W2, thus I would like to raise my score to 3.

**Limitations:**

See the weaknesses

**Paper Formatting Concerns:**

No Formatting Concerns

**Quality:**

2

**Strengths And Weaknesses:**

Strengths:

(1) The paper is well-written and easy to understand.

(2) The theoretical derivation of the paper is sound and clearly organized.

Weaknesses:

(1) The primary concern lies in the motivation of the task itself. Rather than resisting editing effects, the more fundamental and meaningful challenge in defending against malicious editing is to automatically identify which edits are likely to produce harmful outcomes. In fact, resistance can be implemented in a much simpler and more straightforward manner—for example, by directly disabling the editing optimization process, which I believe would offer a more complete form of protection.

(2) In addition, the paper does not provide a clear definition of what constitutes a “malicious” edit, and the visual examples shown in the paper are ambiguous. For instance, editing a subject to resemble an “Asiatic black bear” may not intuitively be perceived as malicious. As mentioned in point (1), I believe the real challenge in defending against malicious editing lies in defining and identifying malicious intentions, rather than simply mitigating their visual effects.

(3) The paper also lacks a clear definition or metric of successful resistance. What is the expected effect achieved by a successful resistance? Why is a straightforward solution like directly stopping the editing process considered insufficient? These questions should be addressed to clarify the necessity and contribution of the proposed method.

---

> ### Author Rebuttal · Authors · 2025-07-30
>
> Thank you for your thoughtful feedback. We have provided point-by-point responses to your concerns as follows.
>
> **Weakness 1: Why not identify harmful edits or simply disable the editing process?**
>
> **Answer:** We would like to clarify that the **(1) core motivation of our task** is to empower content owners to proactively protect their personal 3D digital assets from unauthorized or malicious editing by others. For instance, 3D avatars shared on social platforms can be arbitrarily modified by others, resulting in risks such as impersonation or reputational damage. By applying our protection, the protected 3D assets become resistant to unauthorized modifications, ensuring its authenticity and integrity of 3D avatars in above scenarios. This represents a **user-centric defense paradigm**, enabling asset owners to directly safeguard their content at the source.
>
> **(2)** In contrast, intent-based approaches represent a **platform-centric protection paradigm** that focuses on preventing the occurrence of malicious editing operations. Such methods typically rely on platform or model operators to monitor and regulate user activities. For example, the platform is responsible for monitoring user activity and can directly stop editing operations when malicious editing behavior is detected. While intent-based moderation is valuable for overall platform governance, it does not address the content owner’s need for proactive self-protection.
>
> **(3)** Our task and intent-based methods represent **fundamentally different perspectives**: we focus on user empowerment and protection from the **perspective of content owner**, whereas intent-based solutions approach the problem from the **perspective of platform** monitoring and intervention.
>
> **(4)** In addition, we would like to emphasize that resisting or disrupting malicious editing effects via adversarial noise is **widely recognized** in current research, especially in the area of 2D image editing protection (e.g., AdvDM (ICML’2023) [1], PhotoGuard (ICML’2023) [2], EditShild (ECCV’2024) [3], FaceLock (CVPR’2025) [4], and DiffusionGuard (ICLR’2025) [5]). **Our DEGauss** follows this well-established paradigm and, for the first time, extends it to the emerging domain of 3D editing. As 3D content takes on an essential role in social interaction and digital communication, robust protection of 3D assets is becoming ever more crucial. Our research provides an important foundation for achieving this goal and directly addresses the evolving security needs of generative AI systems.
>
> [1] Adversarial Example Does Good: Preventing Painting Imitation from Diffusion Models via Adversarial Examples. ICML, 2023.
>
> [2] Raising the Cost of Malicious AI-Powered Image Editing. ICML, 2023.
>
> [3] EditShield: Protecting Unauthorized Image Editing by Instruction-guided Diffusion Models. ECCV, 2024.
>
> [4] Edit Away and My Face Will Not Stay: Personal Biometric Defense against Malicious Generative Editing, CVPR, 2025.
>
> [5] DiffusionGuard: A Robust Defense Against Malicious Diffusion-based Image Editing. ICLR, 2025.
>
> **Weakness 2: Explanation of the definition for “malicious” editing and case examples.**
>
> **Answer:** We follow the widely adopted definition of “malicious” editing in previous works [1,2,3,4,5], which refers to **any unauthorized or harmful modification** to the appearance or semantics of 3D assets. The “Asian black bear” example mentioned in this paper is a simple illustrative case that demonstrates our defense's ability to **resist arbitrary operations** by others. The paper also provides more representative examples, such as “turning a person into someone else” (e.g., modifying facial features to resemble a celebrity) or “adding offensive or damaging attributes” (e.g., making someone appear to be on fire). Such cases can cause misrepresentation, defamation, or privacy violations, and are broadly recognized as malicious in the context of digital avatar protection.
>
> We would also like to clarify that our core motivation is to empower individuals to proactively protect their digital assets from malicious tampering, rather than identifying malicious intent or reviewing user behavior at the model or platform level. For most users, it is impossible to control the intentions or intervene in editing process of others. Therefore, we focus on providing a user-centric defense paradigm that can effectively defend against attacks regardless of the intent. In this way, everyone is enable to safeguard their personal digital 3D assets from unauthorized or harmful modifications.
>
> **Weakness 3: Explanation of how to measure successful resistance.**
>
> **Answer 3:** In this paper, successful resistance is defined as making the editing results deviate from the original editing intention. Specifically, we measure both the visual and quantitative differences between the edited 3D asset and the target editing result to determine the effectiveness of resistance. Quantitative metrics widely used in recent studies [3,4,5] are employed to measure semantic differences, including CLIP, CLIP-T, and CLIP-D.  As demonstrated in Figure 3 and Table 1 of the main paper, our method outperforms existing approaches in both qualitative and quantitative.
>
> Furthermore, in real-world scenarios, asset owners typically do not have permission to access the underlying editing workflow, making it impractical to directly disable the editing process. Therefore, our method, which inject protective perturbations into input images to defend against malicious editing, offers a practical and reliable strategy for asset protection.
>
> **We hope these explanations clear up your confusion. We will provide clearer explanations and analysis in the revised version. If you have any further concerns, we would be pleased to provide more detailed clarification.**

---

> > ### Comment · Reviewer_Uu15 · 2025-08-05
> >
> > I appreciate the detailed response. The reasoning behind W1 and W2 is convincing. I would like to raise my score to 3.

---

> > > ### Author Response · Authors · 2025-08-05
> > >
> > > Thank you for taking the time to review our paper. We are pleased to hear that our rebuttal addressed most of your concerns.

---

### Official Review · Reviewer_gPnN · 2025-07-03

**Clarity:** 2
**Significance:** 3
**Originality:** 3
**Rating:** 5
**Confidence:** 3

**Summary:**

The paper proposes a defense method against malicious 3D editing in gaussian splatting. The method aims to ensure that the perturbations are effective in multiple viewpoints by sampling multiple cameras and focusing on the most challenging ones, and further boost this by maximizing the discrepancy between IP2P features as well as IP2P guidance features. They also propose view-focal gradient fusion, where viewpoints are selected that they are far away from each other to avoid focusing on samples from locally close viewpoints, and assigns weights to viewpoints based on the difficulties. Qualitative and quantitative results show that this strategy results in a strong protection compared to baseline methods that are not dedicated for 3D editing (e.g., AdvDM, Mist, SDS).

**Questions:**

**Questions**
- Is it possible to tweak the noise strength down to make it more stealthy while sacrificing some defense strength?
- What is the wall clock optimization time for each method (baselines and proposed method)?
- Would it be possible to use baseline methods combined with some techniques introduced in this paper such as viewpoint sampling? Is there any ablation done in this regard?


**Minor points**
- Typo in Table 3 caption: "GasussianEditor".
- Typo in Figure 5 caption: "GausussianEditor".

**Ethical Concerns:**

["NO or VERY MINOR ethics concerns only"]

**Final Justification:**

Most of my previous concerns have been addressed by the author rebuttal. I think the paper solves a well-motivated problem, and some of the weaknesses in the manuscript that needs revision are the parts related to clarity (e.g., table presentation can be done much more clearly), and I think these weaknesses can be sufficiently addressed through minor revisions. The results in Table R4 is interesting and reassuring, which can be included in the appendix of the final draft. I think overall the contribution of this work is valuable, and therefore raise my rating to 5.

**Limitations:**

yes

**Quality:**

3

**Strengths And Weaknesses:**

**Strengths**
- The motivation to design a defense method dedicated for 3D gaussian splatting editing is clear.
- The paper proposes a novel method to tackle non-trivial challenges that arise when optimizing adversarial noise for 3D objects such as protection being non-uniform across different viewpoints (wide-view sampling and view-focal weighting).
- Extensive results show that the method results in stronger protection strength compared to baseline methods.
- The proposed method seems to preserve the quality of the original image better thanks to the 3D-specific design components.

**Weaknesses**
- Table 2 lacks a comparator or a baseline and is thus not very straightforward to understand.
- The paper lacks a wall time comparison against baseline methods. Is the proposed method slower or is comparable to other methods?
- Although better compared to baseline methods, the adversarial noise is still quite visible on the defended object. Is it possible to tweak the noise strength down to make it more stealthy while sacrificing some defense strength?

---

> ### Author Rebuttal · Authors · 2025-07-29
>
> Thank you for your insightful comments and for helping us improve our work. We have carefully consider and respond point by point to your comments as follows.
>
> **Weakness 1: Table 2 lacks a baseline.**
>
> **Answer:** Thanks for your valuable comment. We have added the unprotected results and the differences before and after protection into Table 2 to show our attack effects more intuitively and clearly, as shown in Table R1. It is observed that our DEGauss significantly and consistently reduces the CLIP similarity with the normal editing results, which proves the effectiveness and generalization of our method under different editing schemes. We only present results for two scenes due to page limitations. The updated complete table and related discussion will be added in the revised version.
>
> Table R1: Quantitative analysis of generalization. “Ori” and “Ours” refer to the non-protected and our protection results, and “Diff” refers to the difference between “Ori” and “Ours”.
>
> |  |  |  |  |  |  | face |  |  |  |  |  |
> |---|:---:|:---:|:---:|---|:---:|:---:|:---:|---|:---:|:---:|:---:|
> |  |  | CLIP |  |  |  | CLIP-T |  |  |  | CLIP-D |  |
> | Method | Ori | Our $\downarrow$ | Diff $\uparrow$ |  | Ori | Our $\downarrow$ | Diff $\uparrow$ |  | Ori | Our $\downarrow$ | Diff $\uparrow$ |
> | GaussianEditor | 1.000  | 0.886  | 0.114  |  | 0.284  | 0.219  | 0.065  |  | 0.084  | 0.033  | 0.051  |
> | DGE | 1.000  | 0.890  | 0.110  |  | 0.253  | 0.230  | 0.023  |  | 0.089  | 0.050  | 0.039  |
> | DreamCatalyst | 1.000  | 0.896  | 0.104  |  | 0.277  | 0.248  | 0.029  |  | 0.089  | 0.044  | 0.045  |
> | EditSplat | 1.000  | 0.933  | 0.067  |  | 0.248  | 0.236  | 0.012  |  | 0.083  | 0.053  | 0.030  |
> |  |  |  |  |  |  | **bear** |  |  |  |  |  |
> |  |  | CLIP |  |  |  | CLIP-T |  |  |  | CLIP-D |  |
> | Method | Ori | Our $\downarrow$ | Diff $\uparrow$ |  | Ori | Our $\downarrow$ | Diff $\uparrow$ |  | Ori | Our $\downarrow$ | Diff $\uparrow$ |
> | GaussianEditor | 1.000  | 0.895  | 0.105  |  | 0.313  | 0.294  | 0.019  |  | 0.062  | 0.026  | 0.036  |
> | DGE | 1.000  | 0.890  | 0.110  |  | 0.310  | 0.298  | 0.012  |  | 0.071  | 0.023  | 0.048  |
> | DreamCatalyst | 1.000  | 0.917  | 0.083  |  | 0.333  | 0.305  | 0.028  |  | 0.074  | 0.035  | 0.039  |
> | EditSplat | 1.000  | 0.935  | 0.065  |  | 0.312  | 0.309  | 0.003  |  | 0.051  | 0.028  | 0.023  |
>
> **Weakness 2 & Question 2: Comparison of wall time with 2D defense methods.**
>
> **Answer:** Thanks for your helpful suggestion. We have evaluated the wall-clock optimization time for our DEGauss and all 2D defense methods over a complete optimization of 2000 iterations, using a consistent 3D setting for all experiments. As shown in Table R2, the wall-clock time of our method is slightly higher than that of AdvDM, Mist, and SDS, as the multi-view weighting strategy incurs additional computational costs. However, our cost is shorter than AdvPaint, because we only compute gradients using a subset of feature maps rather than the full attention map used in AdvPaint. Although some 2D-based defense methods achieve shorter time in simple settings, they perform worse in multi-view 3D protection (higher CLIP scores). Therefore, our DEGauss framework remains comparable to other methods. We will add the table and related discussion in the revised version.
>
> Table R2: Comparison of wall-clock time and defense performance with 2D defense methods. **Bold** indicates the best result.
>
> |  | AdvDM+3DGS | Mist+3DGS | SDS+3DGS | AdvPaint+3DGS | DEGauss (Ours) |
> |:---:|:---:|:---:|:---:|:---:|:---:|
> | Wall-Clock Time $\downarrow$ | 520.06s | 675.49s | **361.01s** | 959.08s | 893.07s |
> | CLIP $\downarrow$ | 0.9235 | 0.9599 | 0.9707 | 0.8996 | **0.8860** |
>
> **Weakness 3 & Question 3: Is it possible to tweak the noise strength down to make it more stealthy while sacrificing some defense strength?**
>
> **Answer:** Yes, it is possible to improve stealthiness by reducing noise strength. We verified the effect of noise strength on stealthiness by uniformly adjusting the perturbation hyperparameter $\lambda$ (jointly $\lambda_{FD}$ and $\lambda_{GD}$), as shown in Table R3. It can be observed that reducing $\lambda$ improves noise stealthiness (higher PSNR score), while decreasing defense performance (higher CLIP scores). When $\lambda =1e\mbox{-}5$, noise invisibility and defense ability reach a trade-off. We will add this experiment to the revised paper.
>
> Table R3: Noise Stealthiness vs. Defense Strength. **Bold** indicates the best result.
>
> | $\lambda$ | PSNR $\uparrow$ | CLIP $\downarrow$ | CLIP-T $\downarrow$ | CLIP-D $\downarrow$ |
> |---|:---:|:---:|:---:|:---:|
> | $1e\mbox{-}4$ | 27.47  | **0.8147** | **0.2018** | **0.0171** |
> | $1e\mbox{-}5$ | 33.92  | 0.8860  | 0.2193  | 0.0325  |
> | $1e\mbox{-}6$ | **34.80** | 0.9617  | 0.2692  | 0.0555  |
>
> **Question 1: Would it be possible to use baseline methods combined with some techniques introduced in this paper.**
>
> **Answer:** Yes, our View-Focal Gradient Fusion strategy (VFGF), include wide-view sampling and view-focal weighting, can be integrated into the baseline methods to boost their performance, as shown in Table R4. It can be seen that our strategy improves the attack performance of all baseline methods, validating the effectiveness of this strategy in multi-view attacks. However, these enhanced baseline methods still appear inferior to our DEGauss, which further highlights the comprehensive advantage of our method. We will include this ablation study and the corresponding analysis in the revised manuscript.
>
> Table R4: Quantitative comparison of baseline methods with and without the integration of our View-Focal Gradient Fusion (VFGF). **Bold** indicates the best result.
>
> |  |  | AdvDM |+VFGF | | Mist |+VFGF | | SDS |+VFGF | | AdvPaint |+VFGF  | | DEGauss (Ours) |
> |:---:|:---:|:---:|:---:|:---:|:---:|:---:|:---:|:---:|:---:|:---:|:---:|:---:|:---:|:---:|
> | | CLIP $\downarrow$ | 0.9235  | 0.9131  | | 0.9599  | 0.9359  | | 0.9707  | 0.9455  | | 0.8996  | 0.8920  | | **0.8860**  |
> | face | CLIP-T $\downarrow$ | 0.2465  | 0.2399   | | 0.2539  | 0.2206  | | 0.2493  | 0.2314  | | 0.2266  | 0.2205  | | **0.2193**  |
> |  | CLIP-D $\downarrow$ | 0.0710  | 0.0679  | | 0.0828  | 0.0792  | | 0.0863  | 0.0853  | | 0.0355  | 0.0330  | | **0.0325**  |
> | | | | | | | | | | | | | | | |
> | | CLIP $\downarrow$ | 0.9072  | 0.8993  | | 0.9564  | 0.9227  | | 0.9546  | 0.9503  | | 0.9046  | 0.8984  | | **0.8949**  |
> | bear | CLIP-T $\downarrow$ | 0.3057  | 0.2988  | | 0.3106  | 0.3084  | | 0.3107  | 0.3036  | | 0.3011  | 0.2996   | | **0.2940**  |
> |  | CLIP-D $\downarrow$ | 0.0378  | 0.0337  | | 0.0373  | 0.0349  | | 0.0378  | 0.0317  | | 0.0314  | 0.0295 |  | **0.0256**  |
>
> **Minor points: typo.**
>
> **Answer:** We have revised the indicated typo and double checked the manuscript to avoid similar typos in the next version. Many thanks for your careful reading.
>
>
> **Thanks again for your insightful feedback. We will incorporate all relevant experiments and discussions into the revised version as suggested. If you have any further questions, we would be glad to offer further explanation.**

---

> ### Author Response · Authors · 2025-08-08
> **We sincerely hope response from the reviewer before the end of the author-reviewer discussion period.**
>
> Dear Reviewer gPnN,
>
> Thank you for the careful evaluation of our paper. We want to know whether our previous response can relieve or successfully address any of your initial concerns. If not and you have any further questions, please feel free to share with us so that we can use our last chance to address them. We want to thank you again for your valuable time and efforts. We will be more than happy to receive your response and resolve any further problems.

---

> > ### Comment · Reviewer_gPnN · 2025-08-09
> >
> > I sincerely appreciate the authors for their detailed feedback. Most of my previous concerns have been addressed. The results in Table R4 is interesting and reassuring, and I hope the authors can include this in the appendix of the final draft. Additionally, I think the paper solves a well-motivated problem, and some of the weaknesses in the manuscript that needs revision are the parts related to clarity (e.g., table presentation can be done much more clearly), and I think these weaknesses can be sufficiently addressed through minor revisions. Other than that, I think the contribution of this work is strong, therefore I raise my score to 5 (Accept).

---

> > > ### Author Response · Authors · 2025-08-09
> > >
> > > We sincerely appreciate your thoughtful evaluation and are pleased to hear that our rebuttal has addressed most of your concerns. We will include those additional results/comparisons in the final version of the paper. Thank you again for your careful consideration and for raising your rating.

---

### Note · Authors · 2025-08-13

Dear Area Chairs,

We sincerely thank the reviewers and area chairs for their valuable time and constructive feedback.

Following our detailed rebuttal, most of the reviewers’ concerns had been satisfactorily addressed, with **two reviewers (2LZe, gPnN) raised their ratings to 5 (Accept)**, other feedback remained positive, and **no new concerns were raised**. Below, we provide a brief summary of the reviewer-author discussion.

- **Reviewer 2LZe** praised “*This paper introduces a novel and timely contribution*”, “*this is the first work … making a meaningful step*”, “*formulation is sound and coherent*”, and “*evaluation is thorough and well-structured*”. In our rebuttal, we standardized notation, quantified loss roles, and detailed baseline adaptation, which he/she confirmed that our detailed response **addressed most of the concerns** and raised the score to **5 (Accept)**.

- **Reviewer gPnN** recognized “*The motivation… is clear*” and “*Extensive results show… stronger protection*”. In response, we provided additional baseline and runtime comparisons, as well as a trade-off analysis. He/She stated that our detailed feedback **addressed most of previous concerns** and the results is “*interesting and reassuring*”. He/She further affirmed that “*the paper solves a well-motivated problem*” and that “*the contribution of this work is strong*”, and raised the score to **5 (Accept)**.

- **Reviewer Uu15** appreciated “*The paper is well-written and easy to understand*” and “*The theoretical derivation … is sound and clearly organized*”. In our rebuttal, we clarified the task of this defense paradigm, stated the widely accepted definition of “malicious editing”, and provided a detailed explanation of metrics. He/She indicated the detailed responses were “**convincing**” and **raised the score**.

- **Reviewer 2tt7** rated all core criteria (*Quality, Clarity, Significance, and Originality*) as **Good** and acknowledging that “*this paper is the first framework ...*” and “*this paper represents a significant piece of work*”. Regarding the concerns raised, we provided runtime evidence, field-standard rationale, and step-by-step optimization details. He/she did **not raise further concerns** during the discussion.

Once again, we deeply appreciate the reviewers’ and area chairs’ efforts and constructive engagement.

Best regards,

Authors of Submission 2332

---

### Decision · Program_Chairs · 2025-09-17

**Decision:**

Accept (poster)

**Comment:**

The paper proposes DEGauss, a defense framework that disrupts malicious 3D editing for Gaussian splatting. The key idea is to use a view-focal gradient fusion mechanism to emphasize the most difficult views during perturbation optimization, along with a dual discrepancy optimization strategy that maximizes semantic and directional deviations. Strengths highlighted across reviews include novelty (first to address this problem in 3D settings), strong motivation, clear formulation, and extensive experiments showing effective and robust multi-view protection while preserving rendering quality. Concerns raised included unclear notation, missing baseline comparisons, computational cost, definition of malicious edits, and lack of clarity on evaluation metrics. The authors’ rebuttal addressed these with revised notation, added ablations, runtime comparisons, and clarified definitions, which reviewers acknowledged as satisfactory.

During discussion, three reviewers moved from borderline to accept after the rebuttal, citing that the authors provided convincing clarifications and additional experiments that resolved most initial weaknesses. One reviewer maintained concerns about the task framing but raised the score after authors clarified their user-centric motivation and alignment with established defense paradigms in 2D. In the end, all reviewers agreed the paper is technically solid, novel, and addresses an important and timely problem. While clarity and presentation could still be improved, the overall contribution is strong enough to warrant acceptance.